# Antenna Cleaning Is Essential for Precise Behavioral Response to Alarm Pheromone and Nestmate–Non-Nestmate Discrimination in Japanese Carpenter Ants (*Camponotus japonicus*)

**DOI:** 10.3390/insects12090773

**Published:** 2021-08-28

**Authors:** Hitomi Mizutani, Kazuhiro Tagai, Shunya Habe, Yasuharu Takaku, Tatsuya Uebi, Toshifumi Kimura, Takahiko Hariyama, Mamiko Ozaki

**Affiliations:** 1Department of Biology, Faculty of Science, Kobe University, Nada-ku, Kobe 657-8501, Japan; morio.24432@gmail.com; 2School of Human Science and Environment, University of Hyogo, Himeji, Hyogo 670-0092, Japan; otakai9813@gmail.com (K.T.); kimura@shse.u-hyogo.ac.jp (T.K.); 3Department of Biotechnology, Graduate School of Science and Technology, Kyoto Institute of Technology, Ukyo-ku, Kyoto 616-8354, Japan; kutibue029@gmail.com; 4Preeminent Medical Photonics Education and Research Center, Institute for NanoSuit Research & NanoSuit Inc., Hamamatsu University School of Medicine, Higashi-ku, Hamamatsu 431-3192, Japan; ytakaku@hama-med.ac.jp (Y.T.); hariyama@hama-med.ac.jp (T.H.); 5*KYOUSEI* Science Center for Life and Nature, Nara Women’s University, Nara 630-8263, Japan; tuebi6@cc.nara-wu.ac.jp; 6Department of Chemical Science and Engineering, Graduate School of Engineering, Kobe University, Nada-ku, Kobe 657-8501, Japan; 7Morphogenetic Signaling Team, RIKEN Center for Biosystems Dynamics Research, Chuo-ku, Kobe 650-0047, Japan

**Keywords:** ant, glooming, antennae, cleaning apparatus, alarm response, nestmate recognition, NanoSuit

## Abstract

**Simple Summary:**

Grooming is a common behavior in animals. It serves the function of removing foreign materials and excessive amounts of self-secreted materials from the body’s surface. Social insects, such as honeybees or ants, use various types of pheromones, some of which propagate information about the environment to conspecific individuals, for chemical communication. The individuals that receive such information can respond with suitable behaviors to protect themselves and their society. Hence, grooming is important for the maintenance of the correct performance of their sensory organs on antennae for pheromone perception. Here, we experimentally limited self-grooming of the antennae in workers of the Japanese carpenter ant (*Camponotus japonicus*) by removing a pair of antennal cleaning apparatuses from the forelegs and investigated their behavioral change in response to exposure to the alarm pheromone or to encounters with nestmates or non-nestmates. Comparisons between self-grooming-nonlimited and self-grooming-limited ants showed that the self-grooming-limited ants gradually exhibited decreased locomotion activity in their fight-or-flight response to the alarm pheromone and experienced increased failure in nestmate and non-nestmate discrimination. Thus, the results of the present study suggest that antennal sensory system maintenance supports social communication, which is indispensable not only to the individual workers but also to the survival of their society.

**Abstract:**

Self-grooming of the antennae is frequently observed in ants. This antennal maintenance behavior is presumed to be essential for effective chemical communication but, to our knowledge, this has not yet been well studied. When we removed the antenna-cleaning apparatuses of the Japanese carpenter ant (*C. japonicus*) to limit the self-grooming of the antennae, the worker ants demonstrated the self-grooming gesture as usual, but the antennal surface could not be sufficiently cleaned. By using scanning electron microscopy with NanoSuit, we observed the ants’ antennae for up to 48 h and found that the antennal surfaces gradually became covered with self-secreted surface material. Concurrently, the self-grooming-limited workers gradually lost their behavioral responsiveness to undecane—the alarm pheromone. Indeed, their locomotive response to the alarm pheromone diminished for up to 24 h after the antenna cleaner removal operation. In addition, the self-grooming-limited workers exhibited less frequent aggressive behavior toward non-nestmate workers, and 36 h after the operation, approximately half of the encountered non-nestmate workers were accepted as nestmates. These results suggest that the antennal sensing system is affected by excess surface material; hence, their proper function is prevented until they are cleaned.

## 1. Introduction

Grooming is a common behavior in many terrestrial animals. Its purpose is to remove foreign materials from the body surface [1,2,3,4,5,6] to prevent infection [7,8,9,10,11] and to remove and/or spread self-secreted materials [8,12,13]. Grooming also occurs when an animal is in a stressed state [14,15,16,17].

Self-grooming of animal antennae is important for the olfactory function and olfactory sense-involved behaviors. When cockroaches are unable to groom their antennae, they become significantly less responsive to periplanone-B, a sex pheromone component [18]; thus, antennae grooming may facilitate courtship performance in cockroaches [19]. Antennal grooming may serve a similar function in a wide range of insect species despite their different grooming styles. Sensory organs may need to be regularly groomed to maintain their responsiveness to the environment of various species [20,21,22,23,24,25,26,27].

Ants are social insects that rely heavily on their olfactory sense; hence, they frequently exhibit self-grooming of the antennae with a pair of special cleaners on their front legs to keep their olfactory sensors functional [28,29,30]. In the Japanese carpenter ant, *C. japonicus*, workers use undecane as an alarm pheromone to warn nestmates of an urgent situation. The nestmates that receive the undecane odor exhibit a fight-or-flight response, sometimes rushing to the odor source and at other times escaping the threat. Their locomotion velocity highly fluctuates. The central neural routes in response to alarming information have been investigated in *Camponotus obscuripes* stimulated with undecane [31,32]. *C. japonicus* wears cuticular hydrocarbon (CHC) mixtures of 18 common components, and the mixing ratio is used as a colony-specific chemical sign. Hence, they can discriminate between nestmates and non-nestmates by detecting differences in these mixing ratios with a specific antennal olfactory organ called the basiconic sensillum [33,34]. Considering sexual dimorphism, the morphology, function, and central neural routes from this type of sensillum have been investigated in several *Camponotus* species [35,36,37,38,39]. Therefore, we used *C. japonicus* in the present study and revealed the effects of self-grooming on the above two types of social chemical communications by comparing video recordings of self-grooming-limited and -nonlimited worker behavior.

Technically, we could easily limit self-grooming of antennae in worker ants by manually removing the antenna-cleaner apparatus pairs. As we adopted the NanoSuit method for antennal observation under a scanning electron microscope (SEM), we obtained clear antennal surfaces images together with surface materials without drift by charging.

## 2. Materials and Methods

### 2.1. Insect

Forager workers of the Japanese carpenter ant, *C. japonicus*, were collected from two different colonies found on the campus of Kobe University, Japan, and were kept in separate plastic cases (40 × 32 × 12 cm) at 20–24 ℃ under a 12L12D cycle in the laboratory. The ants were provided with water and food (36 g yolk powder, 185 g sucrose, 28.25 g calcium, 28.25 g whey protein, and 10 g agar/500 mL water [40]) or 1:4 diluted honey twice a week until use in the experiments. For the heterospecific encounters, *C. obscuripes* workers were obtained from a colony on a hillside in Kobe, Japan and kept in the same manner.

### 2.2. Operation

*C. japonicus* worker ants were individually held with forceps and both forelegs were cut with micro-scissors at the femur–tibia joint so that the antenna cleaner apparatus, consisting of the tarsal notches and the tibial spurs, were completely removed; contrarily, in the sham operation, the forelegs were similarly cut, but the antenna cleaner apparatuses were preserved. After the operation, the ants without antenna cleaners exhibited limited antenna grooming because they could not squeeze their antennae with the antenna cleaner, but the sham-operated ants retained this behavior.

### 2.3. NanoSuit Treatment and Scanning Electron Microscopy

Field-emission scanning electron microscopy was performed using a Hitachi S-4800 (Hitachi, Ltd., Tokyo, Japan) at an acceleration voltage of 1.0 or 5.0 kV. The observation chamber was maintained at a vacuum of 10.3–10.6 Pa. Secondary electrons were accumulated by a lower detector within the instrument. Other experimental parameters were as follows: working distance = 8 mm; aperture size = 100 µm; scan speed = 10–15 fps.

The NanoSuit method [41] was used to observe the intact surface structure of the antenna. A living ant was directly attached to the SEM stub using the carbon conductive double-faced adhesive tape. Any pretreatments such as chemical fixation, dehydration, and ultrathin electrically conducting material coating were not necessary. A small amount (ca. 200 µL) of the commercially available NanoSuit solution Type-I (Nisshin EM Co., Ltd., Tokyo, Japan) was dropped on to the surface of the specimen, which was then wiped as much as possible using dry filter paper to remove excess NanoSuit solution. Then, the specimen was introduced into the SEM to construct a NanoSuit membrane by the irradiation of the electron beam and observed.

Thus, the antennae of different individual NanoSuit-treated ants were observed under the SEM before and 24 and 48 h after the operation. Using this method, we took pictures of the antennal surface with the secession material on it without any inconvenient pretreatment processes for sample preparation and drift by charging. During SEM observation, a specimen ant attempted to move antennae; hence, the fine SEM images could be successfully acquired only when the ants were motionless [42,43]. Nevertheless, it was not difficult to have chances to take fine pictures without blur.

### 2.4. Video Recording and Locomotion Analysis

We randomly selected eight workers obtained from the same colony and introduced them to a round arena of a glass dish (12 cm diameter). At the center of the arena, a small piece of glass wool was set 2.5 cm from the base of the dish by hanging it from a plastic strip bridge, which was used as a releaser. Five min after the ants were introduced, we commenced recording their behavior when 10 μL of 1:1000 diluted alarm pheromone of *C. japonicus*, undecane in *n*-hexane, or plain *n*-hexane as a control was gently added to the glass wool using a micro-syringe. For recordings, a CCD camera (CS8420i, Toshiba Teli Corporation, Tokyo, Japan) with a micro lens (VS-LD25, VS Technology Corporation, Tokyo, Japan) was used. Using the “Windows Live Movie Maker” software (Microsoft, Redmond, WA, USA), the ants’ images at 720 × 480 pixels were recorded for 3 min from the start of undecane or *n*-hexane exposure at 29.97 frames per second, and the video data for the first 10 s were used for locomotion analyses. The recording was repeated 10 times at 2, 12, 24, 36, and 48 h after the antenna cleaner removal operation, and at 2, 12, 36, and 48 h after the sham operation.

For the locomotion analyses, the two-dimensional locational data of each of the eight individuals were obtained frame-by-frame from the recordings using “K-Track” which was originally developed as the multiple bees’ tracking program by Kimura et al. (2014) [44] and the plug-in “Manual Tracking” which is free from the image processing software “Fiji”, originally developed by Schindelin et al. (2012) [45].

### 2.5. Aggression Behavior Test

The sham-operated and antenna cleaner removal-operated ants were individually placed into a round glass container (6 cm diameter, 4 cm height) for up to 48 h and were provided with water. Then, a five-round aggression behavior assay was conducted as follows: a single CO_2_-anesthetized ant was placed into a round plastic arena (4 cm diameter, 4 cm height). After the test ant awoke, a conspecific encounter ant of the same or a different colony from the test ant or a heterospecific encounter ant was gently introduced using a toothpick. Their behaviors were observed, and the test ant’s aggressive or nonaggressive behaviors were recorded until five-round interactions had occurred with the encounter ant. This behavior test with different pairs of the test and encounter ants was repeated six times immediately and every 4 h until 48 h after operation. The number of rounds in which the test ants exhibited aggressive (lashing, biting, or abdomen bending for formic acid spraying) or nonaggressive (light antennation or ignored the presence of the encounter ant) behavior was counted.

## 3. Results

### 3.1. Accumulation of Antennal Surface Material

The antennal surfaces observed before the antenna cleaner removal operation or at different times after the operation are shown in Figure 1. The antennal surface before the operation exhibited few debris of secreted material (Figure 1a), but the antennal surfaces 24 (Figure 1b) and 48 h after the operation (Figure 1c) were lightly and then heavily covered with secreted material, respectively. Almost all antennal sensilla were buried under the surface material by 48 h after the operation. At around 24 h after the operation, various levels of secreted material covering the antennal surfaces were observed in different ant specimens, the surface materials of which were accumulated at different speeds. Opaque white debris spots with secreted material on the antennal surface of such a specimen are shown in Figure 2a, whereas Figure 2b is the magnified form of this image, which focuses on a single basiconic sensillum with debris spots. Surface material was present in minimal amounts on the antennal sensilla within a few hours after the antenna cleaner removal operation, as seen before the operation (Figure 1a and Figure 2c). Secretary organs and their orifices are not identified.

### 3.2. Change in Locomotion with the Alarm Pheromone

Based on the ant location images in the recordings, we calculated their locomotion velocities in the presence of undecane or *n*-hexane. The velocity changed over time after the antenna cleaner operation. As shown in Figure 3a, left, the velocity changes in the presence of undecane were significantly greater at 2 (Wilcoxson signed rank test, *p* = 0.039, *n* = 8) or 12 h (Wilcoxson signed rank test, *p* = 0.0078, *n* = 8) rather than at 24, 36, or 48 h after the operation (Wilcoxson signed rank test, *p* > 0.05, *n* = 8). This was not the case in the presence of *n*-hexane (Figure 3a, right). It is shown in Figure 3b where the integrated variations of ant locomotion velocities are plotted against time after the antenna cleaner removal operation. The integrated variation of locomotion velocities is the sum of absolute values of every second change in locomotion velocity; hence, it can be used as an indicator of locomotion velocity fluctuation during the fight-or-flight response. Further experiments, in which sham-operated ants were used instead of the antenna cleaner-removed ants, were conducted 2, 12, 36 and 48 h after the sham operation (Figure 3c). Those integrated variations of locomotion velocities in the presence of undecane diluted in *n*-hexane or plain undecane were compared with those in the antenna cleaner-removed ants. There were no significant differences, except for 48 h after operation. The integrated variations of locomotion velocities in the sham-operated ants exposed either to undecane diluted in *n*-hexane or plain hexane were mostly constant, whereas those in the antenna cleaner-removed ants were significantly higher 2 or 12 h after the operation than at 36 or 48 h after that (Wilcoxson signed rank test, *p* > 0.05, *n* = 8).

In addition, we counted the number of antennation that occurred between the antennae of the ants (repetitive antennae–antennae contact between ants) (see Figure 4a arrow), and the number of antennation/3 min is plotted against time since the antenna cleaner operation in Figure 4b. The antennation frequency in the presence of undecane was significantly higher than in the presence of *n*-hexane at 2 (Wilcoxson signed rank test, *p* = 0.014, n = 8) and 12 h (Wilcoxson signed rank test, *p* = 0.036, n = 8), but not at 24, 36, or 48 h after the operation (Wilcoxson signed rank test, *p* > 0.05, *n* = 8) (Figure 4b). In most cases, the antennation frequencies of the sham-operated ants in the presence of undecane diluted in *n*-hexane or plain *n*-hexane were significantly higher than in the antenna cleaner-removed ants (Wilcoxson signed rank test, *p* < 0.05, *n* = 8) (Figure 4c).

### 3.3. Change of Aggressiveness

We conducted another behavioral assay at various times after the sham or antenna cleaner removal operation. Each test ant encountered six conspecific nestmates, non-nestmates, or heterospecific workers, respectively, with 5 min intervals, and we counted how many times during the five trials the test ant exhibited aggressive or nonaggressive behavior. The results of the sham-operated, i.e., grooming-nonlimited ants, are shown in Figure 5. They were completely nonaggressive toward conspecific nestmates (Figure 5a), whereas they were always aggressive toward non-nestmates (Figure 5b) and heterospecific workers (Figure 5c), regardless of the time since the sham operation. The results of the antenna grooming-limited ants by the antenna cleaner removal operation are shown in Figure 6. They were mostly nonaggressive toward nestmates until 48 h after the operation (Figure 6a). However, their aggressiveness toward non-nestmates gradually decreased; hence, the aggressive and nonaggressive ratio was reversed 48 h after the operation (Figure 6b). Aggressiveness toward heterospecific workers was completely preserved until 8 h after the operation but decreased stepwise after that (Figure 6c). The aggressive and nonaggressive ratio was almost the same 48 h after the operation.

## 4. Discussion

### 4.1. Improper Reaction to Alarm Pheromone

The behavioral plots in Figure 3b and Figure 4b look similar to one another, even though they are based on different behavioral analyses. Both integrated variation of locomotion velocity and antennation frequency, as indicators of an odor-triggering behavioral response to the alarm pheromone undecane, were significantly higher than those in response to the presence of the solvent *n*-hexane 2 and 12 h after the antenna cleaner removal operation. In fact, the sham-operated ants put in an arena mainly exhibited smooth locomotion along the edge, while they increased irregular locomotion with rushing, turning, or stopping in the presence of certain odor stimulus. However, in our experiments of Figure 3b and Figure 4b, a significant difference disappeared 24, 36, and 48 h after the antenna cleaner removal operation, which implied that the antennal olfactory receptor neurons were no longer sufficiently receiving odor stimulus, in this case, the alarm pheromone, undecane, that triggered irregular locomotion. This might be due to the decrease in the functional antennal olfactory sensilla, which gradually became covered by the surface material (Figure 1). Thus, once the total input of an alarm information was reduced to below a certain threshold, the behavioral responses with irregular locomotion were not observed, as with the response to plain *n*-hexane without undecane (Figure 3b and Figure 4b). This occurred approximately 24 h after antenna cleaner removal, when the surface material including self-secreted CHCs had accumulated on the antennae, as shown in Figure 1b (see Appendix A).

In an agitated state caused by alarm pheromone exposure, the workers decide whether to fight or escape. If there is a sufficient number of nestmates together, individuals can decide to fight. Otherwise, if there are many non-nestmates present, they will be more inclined to escape [46,47]. Therefore, antennation must occur frequently to discriminate between nestmates and non-nestmates. The central neural route activated by undecane exposure has not yet been fully revealed, although neuronal responses to undecane have been recorded in the antennal lobe, projection neurons, and mushroom body [31,32]. However, neural information regarding the alarm pheromone is likely shared for high velocity fluctuation for the fight-or-flight response and for the frequent occurrence of antennation, as long as the antennal sensory system remains appropriately functional through self-grooming. This was the case of the sham-operated control ants, which kept constant levels of integrated variation of locomotion velocity (Figure 3c) and higher antennation frequency than the antenna cleaner-removed ants in the presence of undecane diluted in *n*-hexane or plain hexane (Figure 4c). In Figure 3c, the sham-operated ants tend to show larger integrated variation in locomotion velocity than the antenna cleaner-removed ants. However, the differences are not significant, because the sham-operated ants have a wide distribution of those values.

### 4.2. Ambiguous Nestmate–Non-Nestmate Discrimination

By preventing antennal grooming, we demonstrated that large amounts of surface material accumulated on the ungroomed antennae of *C. japonicus*. The surface material mainly consists of CHCs as the nestmate and non-nestmate discrimination pheromone [33]. Probably, the antennal surface material accumulation has negative influence on antennation, which is a behavior exhibited by many social insects, largely in dominance contexts within the nest and in aggressive contexts towards non-nestmates [48].

Since the self-secreted CHCs are used as signals to identify nestmates and non-nestmates, it was not practical for the ants to remove the antennal surface substance with their grooming mechanism completely. Instead, it is likely that self-grooming serves to redistribute an appropriate amount of cuticular lipids, including self-secreted CHCs, rather than to remove them.

The effects of antennae self-grooming on aggressiveness are shown in Figure 5 and Figure 6. The self-grooming-nonlimited ants were fully aggressive toward both conspecific non-nestmates (with only one exception 16 h after the operation) and heterospecific workers, whereas the self-grooming-limited ants exhibited significantly decreased aggressiveness toward conspecific non-nestmates throughout the experimental period, except for at 0 h (Wilcoxson signed rank test, *p* < 0.05, *n* = 6) and toward heterospecific workers only at 40 and 48 h after the antenna cleaner removal operation (Wilcoxson signed rank test, *p* < 0.01, *n* = 6). An appropriate balance of secretion and sweeping or spreading of antennal surface material including self-CHCs is presumed to be essential for precise nestmate–non-nestmate discrimination.

In cockroaches, self-grooming physically removes excessive native cuticular lipids as well as extrinsic chemicals from olfactory sensilla, thus maintaining the insect’s olfactory acuity to all odorants [18]. This may also be the case in the present study, wherein a reduction in aggressiveness tended to be larger toward conspecific non-nestmates than toward heterospecific workers. In particular, 4 and 8 h after the antenna cleaner removal operation, approximately half of the conspecific non-nestmates were accepted as nestmates, whereas heterospecific workers were all still rejected. The reason for this discrepancy is currently unclear. Nevertheless, we expect that this could be related to peripheral adaptation or desensitization to the CHC mixtures and/or the central matching mechanism to a memorized label of the CHC profile [49,50,51], which might be improved by self-CHCs accumulated on the antennae, but differently between conspecific and heterospecific CHCs.

## 5. Conclusions

Insects have evolved sophisticated systems to interact with the environment via multimodal information, developing various designs of sensors, processors, and actuators. Here, we focused on the ant olfactory sensory organ, the antennae, and the olfactory sensilla in the ant (*C. japonicus*). Our results indicated the importance of antennal cleaning maintenance for the highest performance of the olfactory sensors. Indeed, when the self-grooming of the antennae was limited, the correct performance of two types of chemical communication-related social behaviors, i.e., velocity fluctuation as a fight-or-flight response to the alarm pheromone and nestmate and non-nestmate discrimination by colony-specific CHCs, gradually ceased. Although ants possess finely designed sensory machinery, this cannot be fully utilized without self-grooming of the antennae. Thus, antennal maintenance is indispensable for the survival of ants, not only as individuals but also as societies.

## Figures and Tables

**Figure 1 insects-12-00773-f001:**
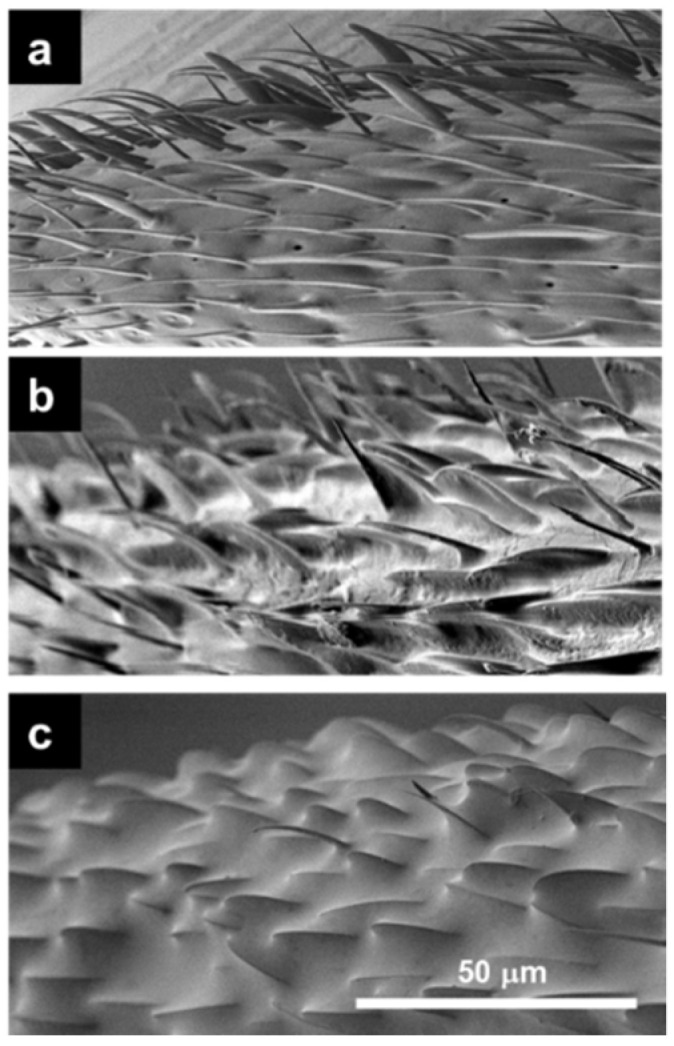
NanoSuit/SEM image of antennal surface. Representative images of antennal surfaces of the tip segments in different individual ants before the antenna cleaner removal operation (**a**), and at 24 (**b**) and 48 h after that (**c**), respectively. The scale is common for (**a**–**c**).

**Figure 2 insects-12-00773-f002:**
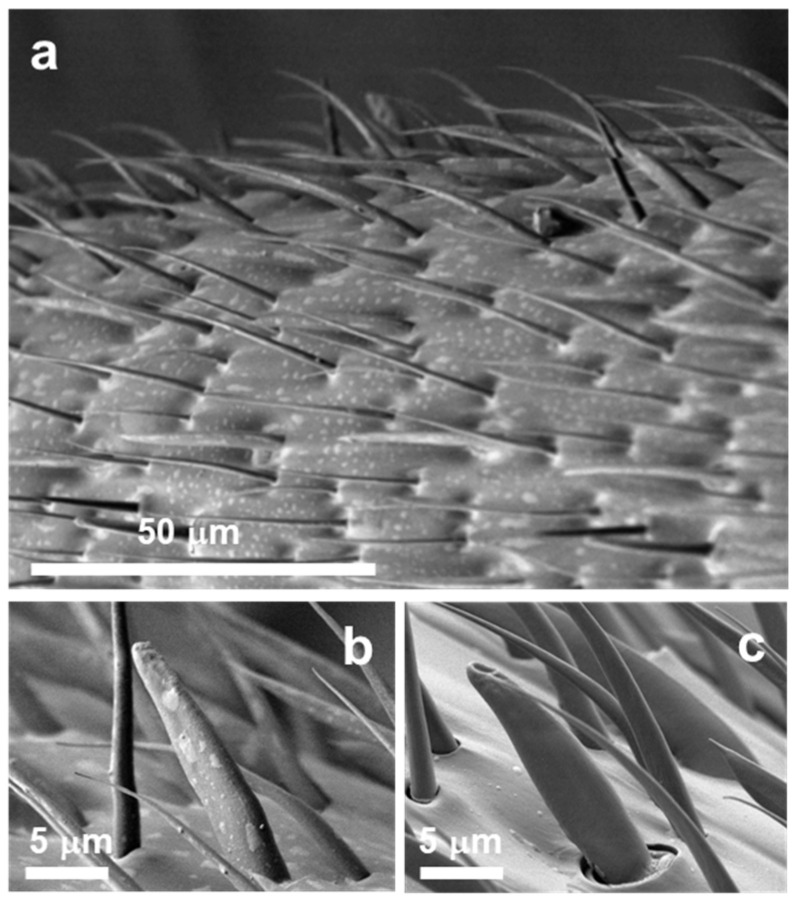
NanoSuit/SEM image of antennal surface and sensillum. Antennal surface (**a**) and a basiconic sensillum in different individual ants at 24 h after the antenna cleaner removal (**b**) and a basiconic sensillum before that (**c**), respectively.

**Figure 3 insects-12-00773-f003:**
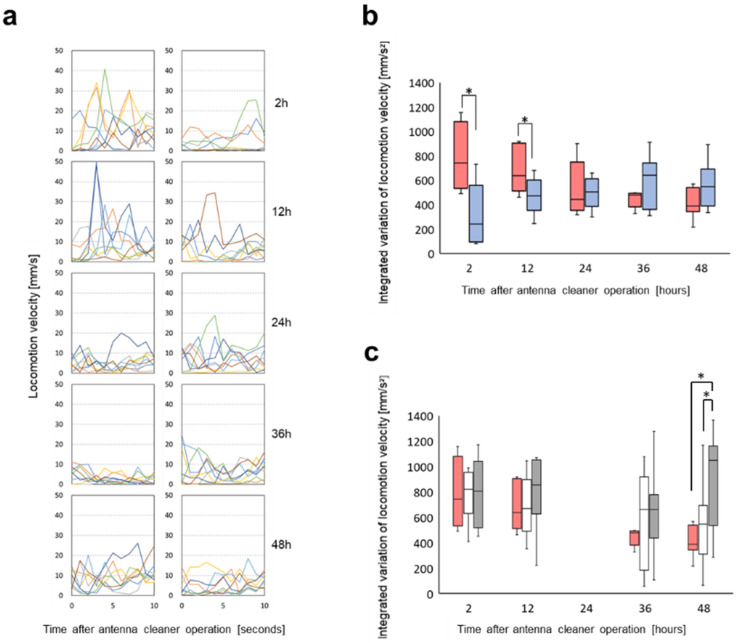
Locomotion velocity changes in the presence of undecane and/or *n*-hexane. (**a**) Locomotion velocity changes in the presence of odor of undecane diluted in *n*-hexane (left) or plain *n*-hexane (right) at 2, 12, 24, 36, and 48 h after the antenna cleaner removal operation (top to bottom). Different colored lines show locomotion velocity changes of eight test ants during 10 s from the beginning of odor exposure. (**b**) The integrated variations of ants’ locomotion velocities are plotted against time after the antenna cleaner removal operation. Red and blue columns indicate the integrated variations of ants’ locomotion velocities in the presence of undecane diluted in *n*-hexane and plain *n*-hexane, respectively. (**c**) Red columns are the same as those in (**b**), but white and gray columns indicate the integrated variations of sham-operated ants’ locomotion velocities in the presence of undecane diluted in *n*-hexane and of plain *n*-hexane, respectively. Asterisks indicate significant difference (Wilcoxon signed rank test, *p* < 0.05, *n* = 8).

**Figure 4 insects-12-00773-f004:**
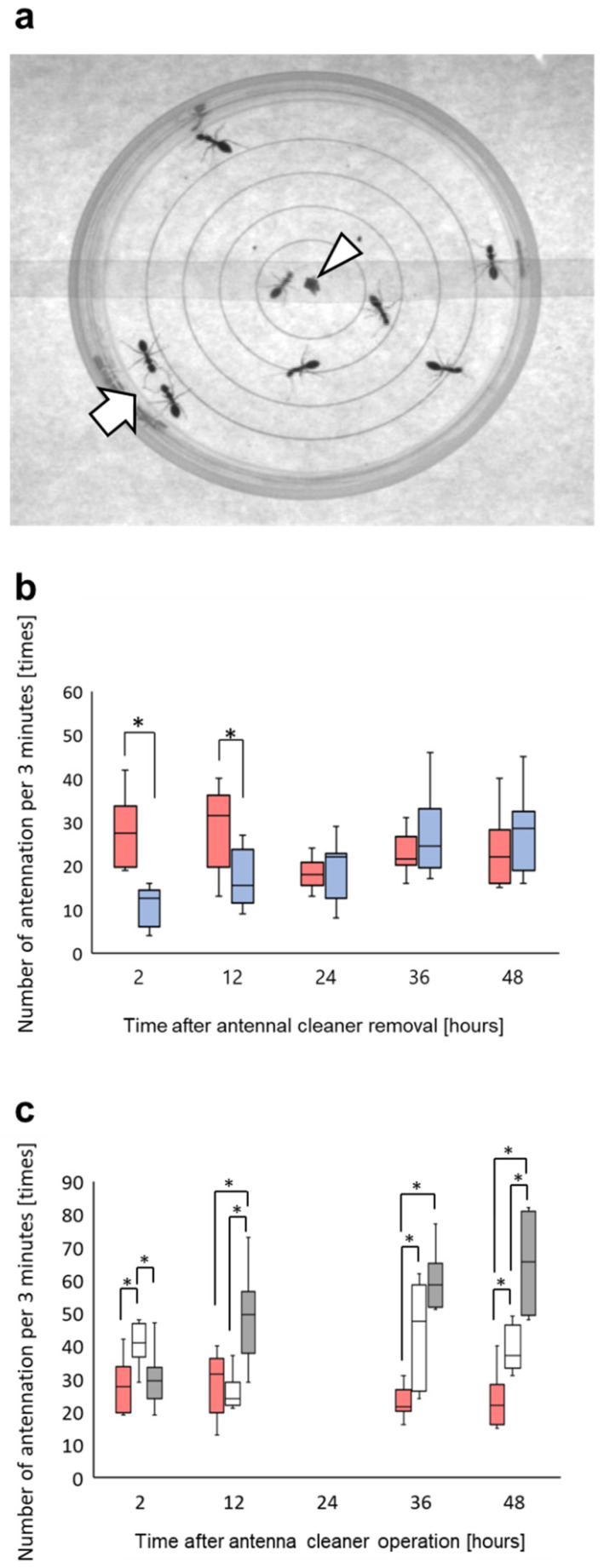
Frequency of antennation between antennae of ants in the presence of undecane and/or *n*-hexane. (**a**) Locomotion arena for video recording. An arrowhead indicates the position of releaser for undecane and/or *n*-hexane. An arrow indicates two individual ants exhibiting antennation between their antennae. (**b**) Such an antennation frequency per 3 min is plotted against time after the antenna cleaner removal operation. Red and blue columns indicate the antennation frequency in the presence of undecane diluted in *n*-hexane and plain *n*-hexane, respectively. (**c**) Red columns are the same as those in (**b**), but white and gray columns indicate the sham-operated ants’ antennation per 3 min in the presence of undecane diluted in *n*-hexane and plain *n*-hexane, respectively. Asterisks indicate significant difference (Wilcoxon signed rank test, *p* < 0.05, *n* = 8).

**Figure 5 insects-12-00773-f005:**
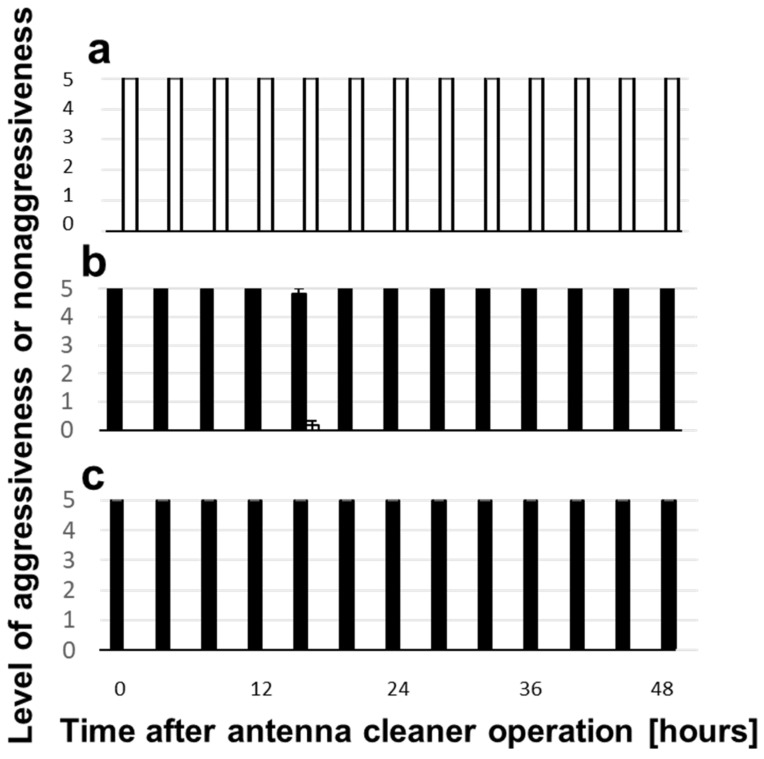
Aggressiveness and nonaggressiveness changes in the self-grooming nonlimited ants. Levels of aggressiveness (black columns) or nonaggressiveness (white columns) in the self-grooming nonlimited ants toward conspecific nestmates (**a**), non-nestmates (**b**), and heterospecific workers (**c**) after the sham operation.

**Figure 6 insects-12-00773-f006:**
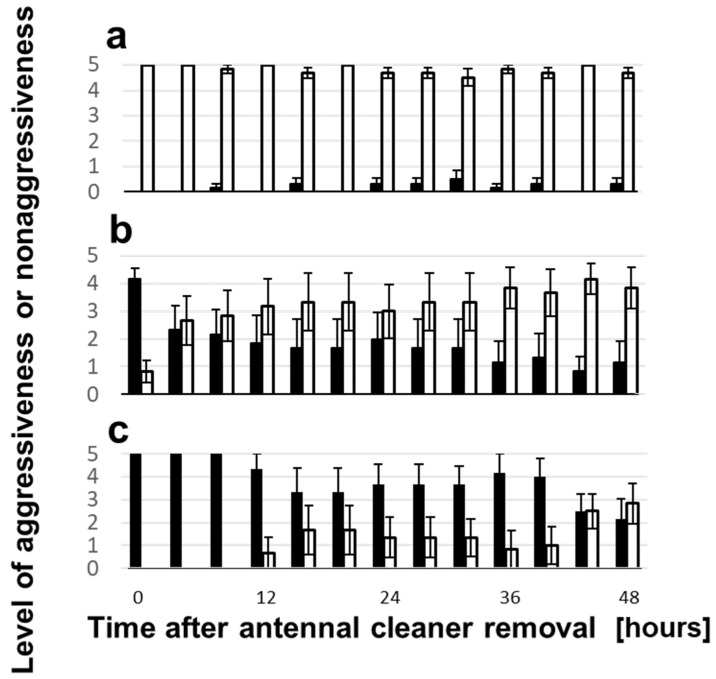
Aggressiveness and nonaggressiveness change in the self-grooming limited ants. Levels of aggressiveness (black columns) or nonaggressiveness (white columns) in the self-grooming limited ants toward conspecific nestmates (**a**), non-nestmates (**b**), and heterospecific workers (**c**) after the antenna cleaner removal operation.

## Data Availability

Data available in a publicly accessible repository.

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
