# Peer review of "Antenna Cleaning Is Essential for Precise Behavioral Response to Alarm Pheromone and Nestmate–Non-Nestmate Discrimination in Japanese Carpenter Ants (Camponotus japonicus)"

_insects, 2021, doi:10.3390/insects12090773_

Round 1
Reviewer 1 Report
Main Comments
The reviewer has read with interest the manuscript titled as “Antenna cleaning is essential for precise behavioral response to the alarm and nestmate-non-nestmate discrimination pheromones in Japanese carpenter ant (Camponotus japonicus)” submitted by H. Mizutani et al. to science journal “Insects”. The authors intend to show with SEM observations and behavioral experiments that excessive accumulation of antennal secreted substances exhibits normal function of antennal pheromone receptors. However there are some serious problems in this manuscript for publication as follows.
- As for the effects of the operation on locomotion shown in Figure 3, the authors compared between the moving speeds of antennal cleaner removed ants exposed to undecane and those of the same operated ants to hexane. However, the effects of antennal cleaner removal on the undecane response should be compared between true operated ants and sham operated ants. Consequently any readers cannot understand whether or not the behavioral changes in shown in Figure 3 are attributable to the antennal cleaner removal because of no control experiments.
- The moving speed of the ants exposed to hexane looks like decrease in Figure 3a but it looks like increase in Figure 3b. What is “integrated variation”? The procedure of integration is not described anywhere in the manuscript. Please show the procedure of the integration and reasons for use of the integrated variation.
- As control data lack in Figure 4b, similar to Figure 3, it is difficult to evaluate whether or not the antennation frequency changes are attributable to the antennal cleaner removal. The authors should compare the frequency of antennation of the true operated ant and that of the sham operated ant.
- As for Finger 6, readers cannot understand whether or not the temporal changes base on the antennal cleaner removal, since it does not show the results of control experiments.
- In introduction the authors told that “Grooming also occurs when an animal is in a stressed state” (L 55). The references (14-16) specialize only vertebrates. Please show some examples of the grooming in stressed states in insects with reference papers.
- Do the authors have evidence for the opinion that the antennal surface material was secreted substance by ants? Please show the grounds with which the authors decided the antennal surface material in Fig 1 as secreted substance of ants.
- The surface materials attach on sensillar shaft as small patches in Figure 2b. Which were the surface materials secreted from on antennal surface? Where are the secretory organs and their orifices
The reviewer does not comment on Discussion at this time.
Minor Comments
L27 apparatus → apparatuses
L41 covered in → covered by
L41 at the same time → concurrently
L66-68 The reviewer thinks this sentence is unnecessary in Introduction
L74 C. japonicus uses cuticular hydrocarbon → C. japonicus wears cuticular hydrocarbon
L86-87 “without the need for any prior inconvenient preparation processes” → This description is unnecessary in Introduction.
L92 cm3 →cm
L103 what behavior is the limited antennal grooming? No explanations on this are found in this manuscript. → Please show clearly the behavior.
L106 and/or → or
L110 (for both SEM instruments) → Did you use two SEMs ?
L112 The whole living ants were stuck directly to the carbon conductive double-faced adhesive tape placed on the SEM stab →Did the stuck living ants move on the SEM tab in a SEM specimen chamber? If the ants moved, how did the authors photograph the ants?
L115 The description manner of instruments and chemicals is not uniformalized. Please show uniformly the product name, maker, maker location.
L133 the Windows Live Movie Maker softwear → Show the name and maker of the software.
L136 Thus → Do the authors think the word “thus” necessary?
L159 Please indicate the stains with arrows in Fig.1a.
Figures 1 and 2 Are the scales shown in these figures correct? The reviewer guesses that scale unit “nm” is an error of “µm“?
Figure 3 Vertical axes in Figure 3a and 3b indicate moving speed and moving velocity, respectively. Are there any differences in meaning between “speed” and “velocity” in this manuscript?
L207 A white arrow in indicates two individual → Omit word “in”.
L227 Consequently →Omit this word.
L264 Unviguous → The reviewer cannot find the word in my dictionaries.
Author Response
The reviewer has read with interest the manuscript titled as “Antenna cleaning is essential for precise behavioral response to the alarm and nestmate-non-nestmate discrimination pheromones in Japanese carpenter ant (Camponotus japonicus)” submitted by H. Mizutani et al. to science journal “Insects”. The authors intend to show with SEM observations and behavioral experiments that excessive accumulation of antennal secreted substances exhibits normal function of antennal pheromone receptors. However there are some serious problems in this manuscript for publication as follows.
As for the effects of the operation on locomotion shown in Figure 3, the authors compared between the moving speeds of antennal cleaner removed ants exposed to undecane and those of the same operated ants to hexane. However, the effects of antennal cleaner removal on the undecane response should be compared between true operated ants and sham operated ants. Consequently any readers cannot understand whether or not the behavioral changes in shown in Figure 3 are attributable to the antennal cleaner removal because of no control experiments.
>We made another comparison experiments between true-operated and sham- operated ants, and the results are shown in Fig. 3c.
・
The moving speed of the ants exposed to hexane looks like decrease in Figure 3a but it looks like increase in Figure 3b. What is “integrated variation”? The procedure of integration is not described anywhere in the manuscript. Please show the procedure of the integration and reasons for use of the integrated variation.
>Figure 3a shows locomotion velocity, while Figure 3b integrated variation of locomotion velocity, i.e., acceleration. We made additional explanation in the text (L214–217 in the revised MS)
As control data lack in Figure 4b, similar to Figure 3, it is difficult to evaluate whether or not the antennation frequency changes are attributable to the antennal cleaner removal. The authors should compare the frequency of antennation of the true operated ant and that of the sham operated ant.
>We made another comparison experiment between true-operated and sham-operated ants, and the results are shown in Fig. 4c.
As for Finger 6, readers cannot understand whether or not the temporal changes base on the antennal cleaner removal, since it does not show the results of control experiments.
>The readers can see the results of control experiments with the sham-operated ants in Fig. 5.
In introduction the authors told that “Grooming also occurs when an animal is in a stressed state” (L 55). The references (14-16) specialize only vertebrates. Please show some examples of the grooming in stressed states in insects with reference papers.
>Reference 16 is not the paper about vertebrate but ant. Anyway, we added one more reference in insects (reference 17).
Do the authors have evidence for the opinion that the antennal surface material was secreted substance by ants? Please show the grounds with which the authors decided the antennal surface material in Fig 1 as secreted substance of ants.
>We added some explanation about the antennal surface material, referring Fig. S1 (New supplementary data of GC analyses of CHCs) (L319–324 in the revised MS).
The surface materials attach on sensillar shaft as small patches in Figure 2b. Which were the surface materials secreted from on antennal surface? Where are the secretory organs and their orifices.
>We answered to this query with the same sentences (L319–324 in the revised MS).
The reviewer does not comment on Discussion at this time.
Minor Comments
L27 apparatus → apparatuses
>We replaced “apparatus” by “apparatuses”.
L41 covered in → covered by
>We replaced “covered in” by “covered by”.
L41 at the same time → concurrently
>We replaced “at the same time” by “concurrently”.
L66-68 The reviewer thinks this sentence is unnecessary in Introduction
>We omitted the sentence in Introduction.
L74 C. japonicus uses cuticular hydrocarbon → C. japonicus wears cuticular hydrocarbon.
>We replaced “C. japonicus uses cuticular hydrocarbon” by “C. japonicus wears cuticular hydrocarbon”.
L86-87 “without the need for any prior inconvenient preparation processes” → This description is unnecessary in Introduction.
> We omitted the sentence “without the need for any prior inconvenient preparation processes” in Introduction.
L92 cm3 →cm
>We replaced “cm3” by “cm”.
L103 what behavior is the limited antennal grooming? No explanations on this are found in this manuscript. → Please show clearly the behavior.
> antenna grooming by squeezing antennae with the antennal cleaner
L106 and/or → or
>We replaced “and/or” by “or”.
L110 (for both SEM instruments) → Did you use two SEMs ?
>We omitted “(for both SEM instruments)”.
L112 The whole living ants were stuck directly to the carbon conductive double-faced adhesive tape placed on the SEM stab →Did the stuck living ants move on the SEM tab in a SEM specimen chamber? If the ants moved, how did the authors photograph the ants?
>We added explanation to answer this question (145–148 in the revised MS).
L115 The description manner of instruments and chemicals is not uniformalized. Please show uniformly the product name, maker, maker location.
>We show uniformly the product name, maker, maker location.
L133 the Windows Live Movie Maker softwear → Show the name and maker of the software.
>We rewrote this part.
L136 Thus → Do the authors think the word “thus” necessary?
>We omitted “thus”.
L159 Please indicate the stains with arrows in Fig.1a.
>No we could not, because there were few visible stains on the antennae before the antennal cleaner removal operation, because the ants freely did the self-grooming with the antennal cleaners.
Figures 1 and 2 Are the scales shown in these figures correct? The reviewer guesses that scale unit “nm” is an error of “µm“?
>We replaced “nm” by “µm“.
Figure 3 Vertical axes in Figure 3a and 3b indicate moving speed and moving velocity, respectively. Are there any differences in meaning between “speed” and “velocity” in this manuscript?
>We revised the vertical axes in Figure 3a and 3b.
L207 A white arrow in indicates two individual → Omit word “in”.
>We omitted “in”.
L227 Consequently →Omit this word.
>We omitted “Consequently”.
L264 Unviguous → The reviewer cannot find the word in my dictionaries.
>We replaced “Unviguous” by “Ambiguous “.
Reviewer 2 Report
In this manuscript the authors apply a straightforward experimental manipulation to answer a simple but very important question regarding the impact that effective self-grooming has on social behaviors that depend heavily on olfactory cues. In my own amply experience using SEM to study social insects I can clearly recognize in their images the impact of secretion accumulation on the outside surface of the cuticle. By applying a technique that allows for the repeated imaging of a given live insect using SEM, the authors are able to record the built-up of secretions and thus of potential “clogging” of the odor receptors in the antennae. Their subsequent behavior essays demonstrate key changes in social response that are very probably due to their experimental manipulation.
In an era of sophisticated techniques for the manipulation of genetic and physiological mechanism, it is nice to be able to read a manuscript that relies on traditional electron microscopy techniques to investigate a relevant question about basic individual behavior (self-grooming) and its impact on colony interaction.
Minor comments:
Line 69. Please remove the extra dot after “Camponotus.”
Figure 1. Please change the “C” in subpanel c to lowercase.
Figures 1 & 2. The scale bar indicates that measurements are given nanometers (“nm”), please confirm if the scale should rather be micrometers (i.e., “um”).
Author Response
In this manuscript the authors apply a straightforward experimental manipulation to answer a simple but very important question regarding the impact that effective self-grooming has on social behaviors that depend heavily on olfactory cues. In my own amply experience using SEM to study social insects I can clearly recognize in their images the impact of secretion accumulation on the outside surface of the cuticle. By applying a technique that allows for the repeated imaging of a given live insect using SEM, the authors are able to record the built-up of secretions and thus of potential “clogging” of the odor receptors in the antennae. Their subsequent behavior essays demonstrate key changes in social response that are very probably due to their experimental manipulation.
In an era of sophisticated techniques for the manipulation of genetic and physiological mechanism, it is nice to be able to read a manuscript that relies on traditional electron microscopy techniques to investigate a relevant question about basic individual behavior (self-grooming) and its impact on colony interaction.
Minor comments:
Line 69. Please remove the extra dot after “Camponotus.”
We omitted the extra dot after “Camponotus.”
Figure 1. Please change the “C” in subpanel c to lowercase.
>We replaced “C” by “c“.
Figures 1 & 2. The scale bar indicates that measurements are given nanometers (“nm”), please confirm if the scale should rather be micrometers (i.e., “um”).
>We replaced “nm” by “µm“.
Reviewer 3 Report
I found this work to be highly significant and worthy of publication. I have made several comments, mostly minor, on the attached manuscript. I suggest you carefully review the manuscript and make changes to make some areas clearer and more easily understandable for the reader. Check the scale bars in the SEM images...I believe they should be in microns. Provide more explanation especially for the data and conclusions for the velocity section. Carefully, review my suggestions included on the manuscript. Overall, an excellent body of work.

Author Response
I found this work to be highly significant and worthy of publication. I have made several comments, mostly minor, on the attached manuscript. I suggest you carefully review the manuscript and make changes to make some areas clearer and more easily understandable for the reader. Check the scale bars in the SEM images...I believe they should be in microns. Provide more explanation especially for the data and conclusions for the velocity section. Carefully, review my suggestions included on the manuscript. Overall, an excellent body of work.
>We replaced “nm” by “µm“.
According to the reviewer #3’s comments on the PDF, we revised our MS as follows;
>We added “under 12L12D cycle”. Ambient temperature was kept by room air-conditioner, although it was not written in the revised MS because I have not seen such a explanation “by room air-conditioner”. (L109 in the revised MS)
>We wrote the species in Abstract. (L51 in the revised MS)
>We replaced “velocity” by “locomotion velocity”. (L87 in the revised MS)
>We replaced “feed” by “food”. (L110 in the revised MS)
>We added citation as reference 39. (L111 in the revised MS)
>We replaced “tweezers” by “forceps” (L117 in the revised MS)
>As “anterior site” has no special meaning, hence we rewrote the sentence to “contrarily in the sham operation, the forelegs were similarly cut but the antenna cleaner apparatuses were preserved.” (L120–121in the revised MS)
>We replaced “as” by “at”. (L127 in the revised MS)
>We omitted “(for both SEM instruments)”. (L131 in the revised MS)
>We rewrote the sentence to “A living ant was directly attached to the SEM stub using the carbon conductive double-faced adhesive tape.” (L133-–134 in the revised MS)
>Yes. See https://www.nisshin-em.co.jp/products/nanosuit%c2%ae%e6%ba%b6%e6%b6%b2
(L136 in the revised MS)
> “Nisshin EM co., Ltd” is the manufacture. (L136-–137 in the revised MS)
>Yes. that is true. (L139-–140 in the revised MS)
>We rewrote the sentence to “Thus, the antennae of different individual NanoSuit-treated ants were observed under the SEM before and 24 and 48 h after the operation, respectively.” (L141—142 in the revised MS)
>We replaced “derived” was replaced by “obtained”. (L151 in the revised MS)
>We replaced ”plastic strip beam” was replaced by “plastic strip bridge”. (L153-–154 in the revised MS)
>We replaced “After the introduced ants appeared calm” by “”Five min after the ants were introduced” (L 154-–155 in the revised MS)
>We replaced the sentence “Using the Windows Live Movie Maker software, the ants’ behavior was recorded for 3 min from the start of undecane or n-hexane exposure at spatial and temporal resolutions of 720 × 480 px/image and 29.97 fps, respectively” by “Using the “Windows Live Movie Maker” software (Microsoft, USA), the ants’ images at 720 × 480 pixels were recorded for 3 min from the start of undecane or n-hexane exposure at 29.97 frame per second,” (L159–162 in the revised MS)
>We omitted “serially” (L166 in the revised MS)
>No, this is not the Windows Live Movie Maker”. (L168 in the revised MS)
>We replaced “regularly-operated“ by antenna cleaner removal-operated” (172 in the revised MS)
> About encountered ant, we added explanation “a conspecific encounter ant of the same or a different colony from the test ant or a heterospecific encounter ant”. (176–177 in the revises MS)
awas repeated six times immediately and every 4 h until 48 h after operation.” (180–182 in the revised MS)
>We replaced “showed” by “exhibited”. (L190 in the revised MS)
> We replaced “stains” by “debris”. (L190 in the revised MS)
>We discussed whether the antennal surface was covered with “secreted material”, referring GC data of the antennae washing in the antenna-grooming-limited (24 h after antenna cleaner removal operation) and nonlimited ants (Fig. S1). (L319–323 in the revised MS)
>We replaced “up to” by “by”. (L193 in the revised MS)
>We replaced “stains” by “debris”. (L199 in the revised MS)
>We replaced “on” by “in minimal amounts on”. (L199–200 in the revised MS)
>We replaced “velocity” by “locomotion velocity”. (L206–207 in the revised MS)
>We corrected the scale in Figs. 1 and 2. Legends were rewritten with “in different individual ants”.
>We rewrote the Y-axis titles of Figs 3a, 3b and 3c (New data) and the related legend and text sentences consistently. (L203–224 in the revised MS)
>We replaced “Singed” by “Signed” in the legend of Fig. 3.
>We added Fig. 4c to compare between the antenna cleaner removed and preserved ants.
>We mentioned the explanation how to measure the level of aggressiveness (0 to 5). (L239–243 in the revised MS)
>We added a citation [31] here. (L278 in the revised MS)
>We replaced “The surface material mainly consists of CHCs working as” by “The surface material mainly consists of CHCs as”, since we disagree with the reviewer’s suggestion “The surface material mainly consists of CHC’s working as”. (L288–289 in the revised MS)
>We added species name. (L329 in the revised MS)
Reviewer 4 Report
This is a complete study showing importance of self-grooming of the antennae on precise behavioral response to alarm pheromone and nestmate-non-nestmate discrimination. The study was well designed and performed. The manuscript was well prepared as well. I only have some minor suggestions.
Title: will it be better to change title as "Antenna cleaning is essential for precise behavioral response to alarm phermone and nestmate-non-nestmate discrimination in Japanese carpenter ant (Camponotus japonicus)"?
Lines 74-77: the sentence is too long. Better to split into at least two.
Line 138: the video recording time is 3 min. Only first 10 s were used for locomotion analyses. Better to specify this here.
Line 179: Do you mean "the number of times of antennation"?
Line 207: delete "in"
Figure 1: are the scales for a, b, c same? If yes, specify in the figure legend. If not, please add scales for a and b.
There are some typos and formating errors in the reference section. Please double-check.
Author Response
This is a complete study showing importance of self-grooming of the antennae on precise behavioral response to alarm pheromone and nestmate-non-nestmate discrimination. The study was well designed and performed. The manuscript was well prepared as well. I only have some minor suggestions.
Title: will it be better to change title as "Antenna cleaning is essential for precise behavioral response to alarm phermone and nestmate-non-nestmate discrimination in Japanese carpenter ant (Camponotus japonicus)"?
>We changed the title according to the reviewer #4.
Lines 74-77: the sentence is too long. Better to split into at least two.
>We split this long sentence into two sentences.
Line 138: the video recording time is 3 min. Only first 10 s were used for locomotion analyses. Better to specify this here.
>We wrote that “the imaging data for the first 10 s were used for locomotion analyses” here.
Line 179: Do you mean "the number of times of antennation"?
>We counted how many times antennation were seen and that number was meant by "the number of times of antennation".
Line 207: delete "in"
>We omitted “in”.
Figure 1: are the scales for a, b, c same? If yes, specify in the figure legend. If not, please add scales for a and b.
> The scales are same for a, b, c. We specify that in the legend of Fig. 1.
There are some typos and formating errors in the reference section. Please double-check.
>We checked and corrected the errors in the reference section.
Round 2
Reviewer 1 Report
The reviewer has read with interest the revised manuscript entitled as “Antenna cleaning is essential for precise behavioral response to alarm pheromone and nestmate-non-nestmate discrimination in Japanese carpenter ant (Camponotus japonica)”, whose title was somewhat changed from the first version, submitted by Mizutani et al. to “INSECTS”. The reviewer read carefully the revised manuscript and recognized that the authors had made widely efforts to improve the manuscript. Two added figures (Figure 3c and Figure 4c) show the results of control experiments, which contain important information in considering the meaning of the antennal cleaning. However some problems remain still now.
Figure 3b and Lines 182-200
Integrated variations of the real operated ants (antennal cleaner removed ants) to undecane decrease along with time but those to hexane increase and after 48h the variations reach almost same level in Figure 3b. Although the reviewer hardly understands the sentence of lines 264-270 in the text, does a certain level of integrated variations exist corresponding to certain insufficient antennal sensory inputs? If so, what do such behaviors mean?
Figure 3b
The integrated variations of sham operated ants to hexane are not shown anywhere. Please show these.
Figure 3c
This figure shows both the results of the real operated ants and the sham operated ants. This figure indicates that no significant differences of the integrated variations are present between the real operated ants and the sham operated ants in any time after the real and sham antennal cleaner removal operations. The result seems to mean that the existence or non-existence of antennal cleaner has no influences on the variation of locomotion velocity. How do the authors think this?
Figure 4b
Similar to the comments on Figure 3b, The results on the frequency of antennation in the presence of hexane are not shown anywhere. Please show these.
As for the vertical axis label, is “frequency” correct? Or does the vertical axis show “number of anntennation”? Please show the dimension (unit). Times is not an adequate unit for frequency.
Figure 4c
It is good that this figure shows the frequency of antennation of both the real and sham operated ants. The antennation frequency is higher in sham operated ants than in real operated ants all times after these operation except 12h after operation. The results probably mean that the antennal substance accumulation has negative influence on the antennation frequency. In this context please show the references on the assumption that the antennation occurs with high frequency in normal condition of antennal chemoreceptors.
Figure 5 and Figure 6
Figures 5 and 6 were the same in size in the first version but are different in size in this version. The reviewer thinks the same size is better.
The reviewer thinks that the horizontal axis label is somewhat inadequate. “time after antennal cleaner operation” may be adequate, because the graph shows the results about both real operated and sham operated ants.
Line 27 apparatus > apparatuses
Line 119 respectively > Is it necessary?
Line 122 a specimen ant would try to move antennae > a specimen ant attempted to move antennae
Line 136 frame > frames
Line 161 by using the NanoSuit-SEM technique > Is this phrase necessary?
Line 192 changes in locomotion velocity or acceleration > Is the phrase “or acceleration” necessary?
Line 201 the number of times antennation occurred > the number of antennation occurred
Lines 223-224 pheromone releaser > releaser (Reason: Hexane is not pheromone)
Lines 273-276 Please show the references related to this part of explanation.
Line 277-280 The reviewer thinks that it may be inadequate to practice this discussion in a relevant way to the reference 31
Lines 302-306 Is this part of description necessary? It may be better that the authors simply state that secretary organs and their orifices are not identified.
Reviewer 4 Report
I am statisified with the revisions to the manuscript.
Author Response
We appreciate the reviewer.
I am glad to here that she/he are satisfied with our revision.